# Distinct Gene Expression Profiles of Matched Primary and Metastatic Triple-Negative Breast Cancers

**DOI:** 10.3390/cancers14102447

**Published:** 2022-05-16

**Authors:** Jaspreet Kaur, Darshan S. Chandrashekar, Zsuzsanna Varga, Bettina Sobottka, Emiel Janssen, Jeanne Kowalski, Umay Kiraz, Sooryanarayana Varambally, Ritu Aneja

**Affiliations:** 1Department of Biology, Georgia State University, Atlanta, GA 30303, USA; jkaur2@student.gsu.edu; 2Department of Pathology-Molecular and Cellular, University of Alabama at Birmingham, Birmingham, AL 35233, USA; dshimogachandrasheka@uabmc.edu (D.S.C.); svarambally@uabmc.edu (S.V.); 3Department of Pathology and Molecular Pathology, University Hospital Zurich, 8091 Zurich, Switzerland; zsuzsanna.varga@usz.ch (Z.V.); annabettina.sobottka-brillout@usz.ch (B.S.); 4Department of Pathology, Stavanger University Hospital, Health Stavanger HF, P.O. Box 8100, 4068 Stavanger, Norway; emiel.janssen@uis.no (E.J.); umay.kiraz@sus.no (U.K.); 5Department of Chemistry, Bioscience and Environmental Engineering, University of Stavanger, P.O. Box 8600, 4036 Stavanger, Norway; 6Livestrong Cancer Institutes, Dell Medical School, The University of Texas at Austin, Austin, TX 78712, USA; jeanne.kowalski@austin.utexas.edu; 7Department of Clinical and Diagnostics Sciences, School of Health Professions, University of Alabama at Birmingham, Birmingham, AL 35294, USA

**Keywords:** triple-negative breast cancer, metastases, RNA-seq, DEGs

## Abstract

**Simple Summary:**

Triple Negative Breast Cancer (TNBC) is a molecularly complex and heterogeneous subtype of breast cancer, characterized by the lack of expression of estrogen receptor, progesterone receptor and human epidermal growth factor receptor 2. TNBCs are often associated with an increased risk of metastasis and recurrence, however, the molecular mechanisms underlying TNBC metastasis and recurrence remail unclear. In this study, we present our findings of massively parallel RNA sequencing used to compare global gene expression profiles of primary tumors and their matched metastatic lesions. Our results shed light on the diverse genetic mechanisms underlying TNBC metastases and may provide potentially actionable therapeutic targets.

**Abstract:**

Background: Although triple-negative breast cancer (TNBC) is associated with an increased risk of recurrence and metastasis, the molecular mechanisms underlying metastasis in TNBC remain unknown. To identify transcriptional changes and genes regulating metastatic progression in TNBC, we compared the transcriptomic profiles of primary and matched metastatic tumors using massively parallel RNA sequencing. Methods: We performed gene expression profiling using formalin-fixed paraffin-embedded (FFPE) TNBC tissues of patients from two cohorts: the Zurich cohort (*n* = 31) and the Stavanger cohort (*n* = 5). Among the 31 patients in the Zurich cohort, 18 had primary TNBC tumors that did not metastasize, and 13 had primary tumors that metastasized (11 paired primary and locoregional recurrences). The Stavanger cohort included five matched primary and metastatic TNBC tumors. Significantly differentially expressed genes (DEGs; absolute fold change ≥2, *p* < 0.05) were identified and subjected to functional analyses. We investigated if there was any overlap between DEGs from both the cohorts with epithelial-to-mesenchymal-to-amoeboid transition (EMAT) gene signature. xCell was used to estimate relative fractions of 64 immune and stromal cell types in each RNA-seq sample. Results: In the Zurich cohort, we identified 1624 DEGs between primary TNBC tumors and matched metastatic lesions. xCell analysis revealed a significantly higher immune scores for metastatic lesions compared to paired primary tumors in the Zurich cohort. We also found significant upregulation of three MammaPrint signature genes (*HRASLS*, *TGFB3* and *RASSF7*) in primary tumors that metastasized compared to primary tumors that remained metastasis-free. In the Stavanger cohort, we identified 818 DEGs between primary tumors and matched metastatic lesions. No significant differences in xCell immune scores were observed. We found that 21 and 14 DEGs from Zurich and Stavanger cohort, respectively, overlapped with the EMAT gene signature. In both cohorts, genes belonging to the MMP, FGF, and PDGFR families were upregulated in primary tumors compared to matched metastatic lesions. Conclusions: Our results suggest that distinct gene expression patterns exist between primary TNBCs and matched metastatic tumors. Further studies are warranted to explore whether these discrete expression profiles underlie or result from disease status.

## 1. Introduction

Triple-negative breast cancer (TNBC) accounts for 15–20% of all breast cancers and exhibits a unique molecular profile [1,2]. TNBC is an aggressive subtype of breast cancer with a higher tendency to metastasize and relapse than other molecular subtypes [3,4]. The high disease heterogeneity and lack of expression of therapeutic targets, including hormone receptors and human epidermal growth factor receptor 2 (HER2), render the treatment of TNBC challenging [4,5]. Additionally, the high rates of tumor recurrence and metastatic progression in TNBC contribute to the high breast cancer–related mortality. Hence, an in-depth understanding of the molecular mechanisms underlying metastatic progression may improve survival outcomes in patients with TNBC.

Metastasis involves the dissemination and spread of cancer cells from the primary tumor site to distant organs, resulting in the formation of treatment-resistant clones capable of proliferation in multiple locations and hindering the success of the treatment [6,7,8]. Successful metastatic dissemination of cancer cells requires careful orchestration of biological events and sequential mobilization of relevant gene expression pathways [7,8]. Hence, elucidating the molecular circuits that regulate metastasis may lead to the identification of prognostic and diagnostic markers as well as therapeutic targets. In turn, the identification of biomarkers and therapeutic targets can help develop detection methods and effective interventions for patients who have not yet developed clinically detectable metastases and for those with advanced disease. In addition to elucidating the underlying genetic programs that drive cancer cell metastasis, understanding the timing of initiation of tumor cell invasion is also critical for effective clinical management of breast cancer.

The aim of this study was to identify genes and pathways regulating the metastatic progression of TNBC and to compare the global gene expression between primary tumors and their matched metastatic lesions. Understanding the molecular alterations associated with metastatic progression may help identify actionable therapeutic targets in TNBC, which are urgently needed.

## 2. Methods

### 2.1. Patients and Patient Samples

Formalin-fixed paraffin-embedded (FFPE) primary and matched metastatic tumor tissues from 36 patients with TNBC were obtained from the University Hospital Zurich, Switzerland, and Stavanger University Hospital, Stavanger, Norway (Table 1). The Zurich cohort was composed of 31 patients, 18 of whom had primary TNBC that did not metastasize, and 13 had primary tumors that metastasized (11 paired primary and locoregional recurrences). In the Zurich cohort, 9 of the 11 (~82%) metastatic or recurrent tumors had metastasized to the lymph nodes, and 2 (18%) were soft tissue and intramammary recurrences. The median age at diagnosis was 56 years. The median follow-up time was 3 years for patients with metastasis and 2.4 years for patients without metastasis. The Stavanger cohort consisted of five patients with primary TNBC tumors that had metastasized to the distant organs—lung, liver, and thorax wall (five paired samples). The median age at diagnosis was 52 years, and the median follow-up time was 2.3 years. The clinicopathological characteristics of all patients are summarized in Table 1. The status of estrogen receptor (ER), progesterone receptor (PR), and HER2 was evaluated using immunohistochemistry (IHC). The study protocol was approved by every Institutional Review Board and was in compliance with material transfer guidelines and data use agreements between Georgia State University and the participating institutes. This study was conducted in accordance with International Ethical Guidelines for Biomedical Research involving human subjects. Written informed consent was obtained from all participants in the Zurich cohort. For the Stavanger cohort, the Regional Ethical Committee gave permission to use these samples and patient records without consent as this was a retrospective cohort and all patients had died by the time of this study.

### 2.2. RNA Isolation and Sequencing

Hematoxylin and eosin (H&E) slides were prepared for all samples, and the tumor content was assessed by a pathologist. Total RNA was extracted from FFPE tissue sections using a NucleoSpin total RNA FFPE kit (Macherey–Nagel; Düren, Germany). RNA samples were subjected to optical density measurements using NanoDrop and Qubit (Thermo Fisher Scientific; Waltham, MA, USA). RNA purity and concentration were determined using 2200 Tapestation (Agilent Technologies, Inc.; Santa Clara, CA, USA). RNA integrity number (RIN) according to Agilent 2100 assays depend on sample type and quality. In general, quality of RNA extracted from FFPE samples is poor. Typical RIN of RNA isolated with NucleoSpin RNA FFPE kits (Macherey–Nagel; Düren, Germany) are in the range of 2–6. The RIN of total RNA isolated from FFPEs for most of the samples (87%) was 2 or above and for a few samples was ~2 (Appendix A). Illumina sequencing libraries were prepared using the TruSeq Stranded mRNA kit (Illumina; Mountain View, CA, USA) following the manufacturer’s instructions. The resulting library was tested for size distribution and concentration using 2200 Tapestation, Nanodrop, and Qubit. The libraries were sequenced on a NextSeq 500 (Illumina; San Diego, CA, USA) according to the standard operation. Paired-end, 150-nucleotide reads were generated, and data quality was assessed using FASTQC (Babraham Institute, Cambridge, UK).

### 2.3. RNA-Seq Data Processing

Raw FASTQ files were subjected to quality control analysis using FASTQC [9]. Raw sequencing reads were trimmed to remove adapter sequences and low-complexity regions using Trim-Galore. Trimmed reads were mapped to the human reference genome GRCh38 using TopHat2 [10]. Total paired end reads and % reads mapped to the human reference genome are reported in the Appendix A. Mapped reads were sorted using SAMtools [11], and HTSeq [12] was used to obtain raw read counts for each gene. Differential gene expression analysis was performed using DESeq2 as described previously [13]. We identified significant differentially expressed genes (DEGs) by adjusting the absolute fold change to ≥2 and a *p*-value to <0.05. Significant DEGs were used to generate heatmaps and volcano plots. Gene ontology (GO) and Kyoto Encyclopedia of Genes and Genomes (KEGG) enrichment analyses were used to predict the biological roles of the DEGs. This analysis was performed using the Database for Annotation, Visualization, and Integrated Discovery (DAVID) version 6.8, which is a web-based functional annotation tool. We used web-based Venny 2.1 tool to generate the Venn diagrams [14]. The list of genes in EMAT-related gene signatures is provided in Appendix A [15].

### 2.4. Cell Type Enrichment Analysis

Estimation of cell type abundance was performed using the bioinformatics tool xCell and normalized bulk RNA-seq expression data as input. xCell is a high-resolution gene-signature-based method for estimating the tumor’s immune and stromal cell composition [16]. The relative abundance of cell types was quantified and visualized across all samples. Immune and stromal scores were compared across different groups.

## 3. Results

### 3.1. Identification of DEGs between Primary TNBC Tumors and Matched Metastatic Lesions

Differential gene expression analysis was conducted separately for Zurich and Stavanger cohorts. We identified a total of 1624 and 818 DEGs (fold change ≥2, *p* < 0.05) between primary tumors and matched metastatic lesions in the Zurich and Stavanger cohorts, respectively. In the Zurich cohort, 253 genes were upregulated, and 1371 genes were downregulated in primary tumors compared to matched metastatic lesions (Figure 1A). In the Stavanger cohort, 326 genes were upregulated, and 492 were downregulated in primary tumors compared to metastatic lesions (Figure 1B). By comparing the DEGs in the two cohorts, we identified 28 common upregulated and 39 common downregulated genes in primary tumors (Figure 1C,D). Among the common upregulated genes in primary TNBC tumors, we identified genes that have been implicated in cancer cell invasion and migration, including genes belonging to the MMP, PDGF, and FGF gene families [17,18,19,20]. Specifically, *MMP13*, *FGF7P*, and *PDGFR* were upregulated in both Zurich and Stavanger cohorts. The top 100 upregulated and downregulated DEGs (ranked by fold change and *p*-values) between primary tumors and metastatic or recurrent tumors in Zurich and Stavanger cohorts are shown in Figure 2. We also compared the DEGs between primary TNBC tumors that metastasized and those that did not metastasize from the Zurich cohort. In total, 832 genes were upregulated, and 906 genes were downregulated in metastatic primary TNBC tumors compared to non-metastatic primary tumors (Appendix A). Interestingly, *HRASLS*, *RASSF7*, and *TGFB3* (part of the MammaPrint assay) were significantly upregulated in metastatic primary TNBC tumors compared to non-metastatic primary tumors (Appendix A). MammaPrint is a 70-gene signature used to predict tumor recurrence and risk of metastasis in patients with breast cancer [21,22]. To gain deeper insights into the biological functions of DEGs, we performed GO and KEGG pathway enrichment analyses. Using a false discovery rate (FDR) threshold of <0.05, we identified 104 enriched GO biological processes and 28 enriched KEGG signaling pathways in the Zurich cohort (Appendix A); 58 GO biological processes and 17 KEGG signaling pathways were enriched in the Stavanger cohort (Appendix A). Interestingly, most of the top 20 biological processes enriched in the Zurich cohort were related to immune responses, including T-cell/B-cell activation and T-cell/B-cell receptor signaling. Furthermore, DEGs were enriched in cytokine–cytokine receptor interaction, T-cell receptor signaling, and NF-κB signaling.

### 3.2. Analysis of EMAT-Related Gene Signature

EMT plays a key role in development as well as in cancer cell invasion and metastasis [23]. Elucidating the role of EMT in metastasis often involves in vitro or in vivo studies because it is challenging to assess metastatic samples from patients. Only a few validation studies have been conducted using matched pairs of human primary and metastatic samples [24]. Therefore, we investigated if there are any common genes between epithelial-to-mesenchymal-to-amoeboid-transition (EMAT)-related gene signatures (385 genes) and the DEGs between primary and metastatic tumors for both of our cohorts. We found that 21 and 14 differentially expressed genes between primary and paired metastatic lesions from Zurich and Stavanger cohort, respectively, were concurrent with EMAT gene signatures (Figure 3 and Appendix A). Interestingly, of the 21 and 14 DEGs from the two cohorts that intersected with EMAT gene signatures, there were four genes (*PDGFRL*, *UCHL1*, *COL5A2* and *COL3A1*) (Appendix A) that were common between the Zurich and Stavanger cohort. 

### 3.3. Comparison of Immune, Stromal, and Microenvironment Scores in Paired Primary and Metastatic TNBC Samples

The tumor microenvironment (TME) is composed of malignant cells, stromal infiltrates, and immune cells. The complex interplay between these TME components can influence different tumor properties, including cancer cell invasion, metastasis, and therapy resistance [25,26]. To gain further insight into the role of cellular heterogeneity within the TME in TNBC metastasis, we performed xCell analysis for primary tumors and matched metastatic lesions. In the Zurich cohort, the immune score was significantly higher in metastatic lesions than in paired primary tumors; however, no significant differences in the stromal score were observed (Appendix A). Notably, significant differences in the scores for CD4 T cells, CD4 T central memory cells, CD4 T effector memory cells, CD8 T cells, naïve B cells, memory B cells, class-switched memory B cells, and epithelial cells were observed between primary and paired metastatic tumors. The immune landscape across the paired samples is presented in Figure 4. We also compared the immune and stromal scores between the metastatic and non-metastatic primary TNBC tumors (Appendix A). No significant differences were observed in overall immune or stromal scores between the two groups. However, we found significantly higher scores for CD4 T effector memory cells, CD8 T central memory cells, CD8 naïve T cells, dendritic cells, mast cells, megakaryocytes, Th2 cells, gamma delta (TgD) cells, activated dendritic cells (aDCs), conventional dendritic cells (cDCs), and pro-B cells in non-metastatic primary TNBC tumors compared to metastatic primary TNBC. In the Stavanger cohort, no significant differences in immune, stromal, or microenvironment scores were observed between paired tumors (Appendix A).

## 4. Discussion

In this study, we compared the transcriptomic profiles of matched primary and metastatic tumors using RNA-seq to identify molecular changes and DEGs regulating metastasis in TNBC. Primary tumors and matched metastatic tumors showed distinct gene expression patterns as multiple DEGs and differentially regulated pathways and immune components were identified between the two groups. Discordance in the transcriptomic profiles of primary and matched metastatic TNBC tumors can be attributed to temporal and spatial differences between primary and metastatic lesions and the effects of neoadjuvant chemotherapy received by 4 of the 11 (~40%) patients with matched primary and locoregional recurrences (Zurich cohort); neoadjuvant chemotherapy may have led to changes in the gene expression profile of tumors in these patients. These distinct gene expression profiles may indicate a molecularly dynamic tumor adapting to a new microenvironment that supports metastatic selectivity. Comprehensive analyses integrating various omics techniques can be undertaken as further cohorts with paired primary and metastatic tumors become available. We also found genes belonging to the MMP, FGF, and PDGFR families to be commonly upregulated in both the cohorts. MMPs are known to contribute to each step of breast cancer metastatic cascade owing to their ability to cleave various non-matrix and matrix substrates [27,28]. It has been demonstrated that tumor cell–derived FGFs contributed to the formation of metastatic lesions in vivo [29]. Particularly in TNBCs, high PDGFR expression is associated with lymph node metastases and tumor recurrences [20,30]. 

We investigated if there is any overlap between the DEGs from both the cohorts with EMAT gene signature which is implicated in considering both, the EMT and MAT continuum instead of either of the gene signatures to capture the heterogeneity of metastatic propensity. Since study by Emad et al. implicated the true clinical and prognostic significance of EMT as a driving process in cancer progression towards distant metastasis can be fully appreciated if it is complemented by the additional occurrence of MAT, a process that plays an important role in embryonic development and is similarly reawakened (as EMT) by cancers during the metastatic cascade [15,31]. Our results indicate 21 and 14 DEGs from Zurich and Stavanger cohort, respectively, that overlapped with the EMAT gene signatures. Specifically, there were four genes, namely *PDGFRL*, *UCHL1*, *COL5A2,* and *COL3A1* that were common among all three-DEGs from the EMAT gene signature, the Zurich and the Stavanger cohort. Interestingly, all these genes have been implicated in promoting invasion and metastasis and associated with poor survival outcomes in breast cancer patients [32,33,34]. Previous studies also found that the expression levels of *COL3A1* were higher in primary TNBC tumors compared to lymph node metastasis [35]. Role of *UCHL1* in promoting breast cancer progression has been well documented. It has been shown to promote breast cancer metastasis by maintaining TGF-β signaling pathway and promoting breast cancer cells invasion by activating Akt signaling [33,34]. 

We found significant upregulation of *HRASLS*, *TGFB3*, and *RASSF7* in metastatic primary TNBC tumors compared to non-metastatic primary tumors. Interestingly, these genes are part of MammaPrint, a 70-gene signature that is used to predict the risk of recurrence and metastasis in breast cancer [21,22,36]. These genes have been implicated in cancer cell proliferation and immortality [37]. Specifically, *TGFB3* contributes to aggressive phenotypes by promoting invasiveness, angiogenesis and creating an immunosuppressive environment [38,39]. *HRASLS* and *RASSF7*, oncogenic transformation-related genes, have been implicated in contributing to at least three of the hallmarks of cancers (evading apoptosis, self-sufficiency in growth signals, and insensitivity to anti-growth signals) described by Hanahan and Weinberg [37,40]. DEGs between primary tumors with and without metastasis could serve as therapeutic targets to prevent early stages of metastasis or as biomarkers to identify patients who are at low risk of metastasis and, hence, could be spared from unnecessary surgical procedures. However, we do not exclude the possibility that the differences observed between the primary tumors that metastasized and those that did not could be attributed to difference in the follow-up times and histopathological characteristics between the groups and not solely to genomic differences.

Previously, studies have shown that the immune profile of tumor-draining nodes is predictive of survival outcomes in breast cancer patients [41]. xCell analysis used to identify differences in immune and stromal profiles between primary tumors and matched metastatic lesions revealed a significantly higher frequency of CD8 T cells, CD4 cells, and effector/memory CD4 T cells in metastatic tumors, most of which were locoregional recurrences in lymph node. Our finding is in line with previous reports of high levels of these immune cell types in metastatic lymph nodes [42,43]. The presence of high numbers of specific immune cell types may indicate an ongoing immune response since immunity conferred by effector/memory T cell is traditionally considered to be antitumoral [43]. This observation is further supported by the results of pathway analysis, which indicated an enrichment of immune response, T-cell activation, and T-cell receptor signaling pathways. This is in sync with similar results observed by Ellsworth et al., where they found a higher expression of genes associated with immune response in lymph node metastases compared to the primary breast tumors [44]. However, we do not preclude the possibility that the observance of higher immune score or differentially expressed genes in locoregional recurrences compared to primary breast tumors for the Zurich cohort could be in part attributed to the basic nature of the metastatic organ, lymph nodes in this case. Hence, further characterization of the immune and stromal cells through high-dimensional maps of the interactions between tumor cells, immune cells, and stromal cells in different metastatic sites is warranted to provide detailed insights into the role of these interactions in TNBC metastasis.

There are some limitations to this study. First, even though these cohorts represent a unique resource of matched primary and metastatic tumors, their sample sizes were small. As most institutions do not routinely biopsy metastatic or recurrent tumors, obtaining matched sets of primary and metastatic or recurrent tumors is challenging. Second, the cases chosen for this study were based on sample availability, inadvertently introducing selection bias. Third, because this was a retrospective study and some FFPE samples were old, it was difficult to isolate high-quality RNA. Finally, the low number of significant DEGs in the Stavanger cohort could be attributed to the small sample size and may represent false positives.

## 5. Conclusions

Our results demonstrate that metastatic cancer cells undergo a biologically significant transcriptomic shift upon colonization. Further studies are warranted to explore whether gene expression profiles associated with metastasis could serve as actionable therapeutic targets in TNBC.

## Figures and Tables

**Figure 1 cancers-14-02447-f001:**
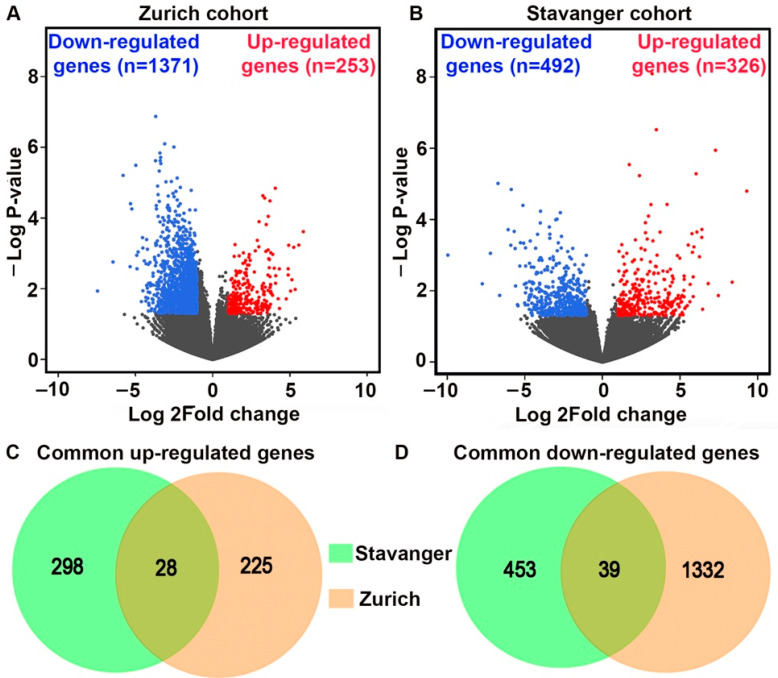
Gene expression profiling and identification of DEGs between primary and metastatic tumors. (**A**,**B**) Volcano plots showing the distribution of DEGs in Zurich and Stavanger cohorts. (**C**,**D**) Venn diagrams depicting genes commonly upregulated or downregulated in primary tumors in Zurich and Stavanger cohorts.

**Figure 2 cancers-14-02447-f002:**
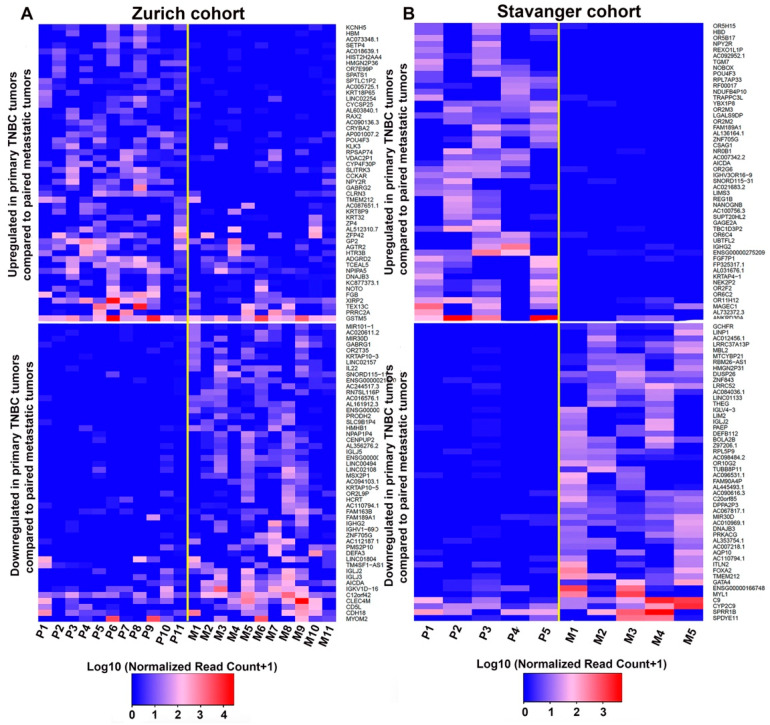
DEGs between primary and metastatic TNBC tumors. (**A**,**B**) Heatmap showing the top 100 upregulated (top) and downregulated (bottom) genes in primary TNBC tumors (*n* = 11) compared to matched metastatic tumors (*n* = 11) in the Zurich cohort (**A**) and in primary TNBC tumors (*n* = 5) compared to matched metastatic tumors (*n* = 5) in the Stavanger cohort (**B**). P = Primary tumor; M = Metastatic tumor.

**Figure 3 cancers-14-02447-f003:**
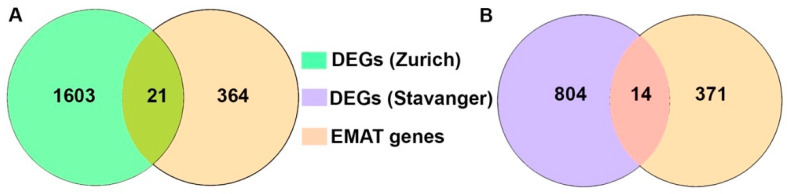
Analysis of the EMAT-related gene signature. Venn diagram depicting the overlapping genes between the EMAT gene signature and DEGs from Zurich (**A**) and Stavanger (**B**) cohorts.

**Figure 4 cancers-14-02447-f004:**
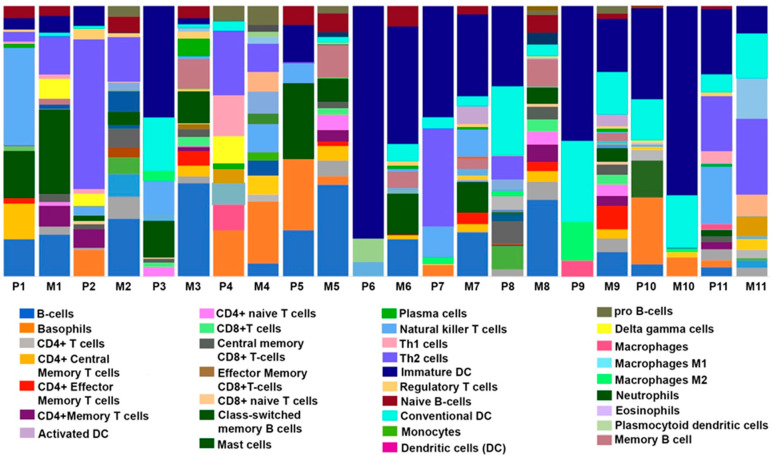
Cellular landscape of immune microenvironment in primary TNBC tumors and matched metastatic lesions. The proportions of immune cells in each TNBC (primary “P”; *n* = 11 or metastatic “M” tumors; *n* = 11) sample are indicated with different colors. The lengths of the bars in the bar charts indicate the levels of the immune cell populations.

**Table 1 cancers-14-02447-t001:** Clinicopathological characteristics of patients with TNBC in Zurich and Stavanger cohorts.

Baseline Characteristics	Metastasis (Lymph Node or Other)	Metastasis-Free	Metastasis (Distant Organ)
Zurich cohort (*n* = 31)			Stavanger Cohort (*n* = 5)
Patient Age, *n* (%)			
20–29	0 (0.00)	1 (5.55)	0 (0.00)
30–39	2 (15.38)	1 (5.55)	1 (20.00)
40–49	1 (7.69)	4 (22.22)	1 (20.00)
50–59	5 (38.46)	6 (33.33)	1 (20.00)
60–69	3 (23.07)	1 (5.55)	2 (40.00)
70+	2 (15.38)	5 (27.78)	0 (0.00)
Tumor Grade, *n* (%)			
I	0 (0.00)	0 (0.00)	0 (0.00)
II	1 (7.69)	2 (11.11)	0 (0.00)
III	12 (92.31)	16 (88.89)	5 (100.00)
Missing	0 (0.00)	0 (0.00)	0 (0.00)
Histological Type, *n* (%)			
NST (ductal)	10 (76.92)	17 (94.44)	5 (100.00)
NST (with secretory differentiation)	1 (7.69)	0 (0.00)	0 (0.00)
Apocrine	1 (7.69)	1 (5.56)	0 (0.00)
Metaplastic	1 (7.69)	0 (0.00)	0 (0.00)
Missing	0 (0.00)	0 (0.00)	0 (0.00)
Survival Status, *n* (%)			
Alive	5 (38.46)	13 (72.22)	0 (0.00)
Dead	8 (61.54)	5 (27.78)	5 (100.00)
Missing	0 (0.00)	0 (0.00)	0 (0.00)

## Data Availability

The data presented in this study are available in this article (and Appendix A).

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
