# Peer review of "Distinct Gene Expression Profiles of Matched Primary and Metastatic Triple-Negative Breast Cancers"

_cancers, 2022, doi:10.3390/cancers14102447_

Round 1
Reviewer 1 Report
According to my point of view there was no improvement to the first version.
I still missed a proteome analysis. At least HRASLS, TGFB3 and RASSF7 expression should be confirmed with Northern-hybridization.
The only advantage of transcriptomic analysis to the proteomic one that can be detected the expression of pseudogenes.
Here there is e.g. VDAC2-P1 over-expression detected. What can be the biological relevance of it?
I would not support this manuscript to be published in Cancers.
Reviewer 2 Report
All review concerns addressed in the revised manuscript.
Reviewer 3 Report
Corrections have been made
I suggest to accept
This manuscript is a resubmission of an earlier submission. The following is a list of the peer review reports and author responses from that submission.
Round 1
Reviewer 1 Report
The manuscript entitled Distinct gene expression profiles of matched primary and metastatic triple-negative breast cancers aimed to identify the transcriptional differences between the primary and metastatic tumors in order to monitor the changes during the metastatic progression in triple-negative breast cancer (TNBC). Therefore the transcriptomic profiles of primary and matched metastatic tumors were compared using massively parallel RNA sequencing.
They investigated 2 cohorts one comprised n=31, and the other consisted of n=5 cases. The authors combined significantly differentially expressed genes (DEGs) to unravel the differences in the genetic regulation of the metastatic progression.
The main flaw of this manuscript is the approach.
In the past the high throughput analysis systems aiming to find any differences in the transcriptomic profiles were rarely successful. Most probably these led the devaluation and neglect of this techniques.
There was a permanent critics against the transcriptomic approach what are the biological relevance of an mRNA without a protein profile.
In this manuscript we can find an analysis of transcriptomic profiles without any success. They found only one gene (TWIST1) known implicated in EMT.
Further they reported three MammaPrint signature genes (HRASLS2, RASSF7, and DTL), that I could not find them in any pictures.
In advance, it was not clear why they expected to see any genes related to EMT in metastatic lesions. EMT means epithelial-mesenchymal transition that is a relative fast process, after the cells migrated and settled there is a so called MET, mesenchymal-epithelial transition. Therefore the missing EMT genes are not a surprising result.
Taken together, the original question what can be the difference between the primary and metastatic tumors is valuable. Instead of the transcriptomic analysis I would suggest a proteomic approach, that is cheaper faster and mostly more relevant to unravel the differnces. Even in the case of small lesions a so called single cell analysis can be used.
Reviewer 2 Report
This manuscript by Jaspreet Kaur et al investigated the differential gene expression profiles of matched primary and metastatic TNBC tumors. The results are of interest, however, there are a few things that can be addressed to make the manuscript even more interesting and impactful. For instance, it will be nice to see if there are any genes that overlap between the two types of metastatic tumors (nodal and distant organ metastatic) that they reported. Authors have done the analysis of EMT related genes between the different comparison groups, however, do not provide a reference for the same. The authors reported that genes belonging to MMP, FGF and PDGFR families to be commonly upregulated in both cohorts. I suggest adding in more details specifically focusing the role of these genes in promoting metastasis. Further, the authors report upregulation of 3 genes namely, HRASLS2, RASSF7 and DTL which are a part of MammaPrint 70 gene signature. It will be a great idea to add in more details about the metastatic role of these genes to enhance the discussion.
Reviewer 3 Report
the authors report a gene expression analysis from paired primary tumors and metastases from triple negative breast cancer
Although interesting, the report suffers from many important flaws, and deserves some important corrections and improvements
- why the two cohorts have not been analysed simultaneously ? this should be explained. Furthermore, it is regularly stated that the Stavanger cohort (n= ...only 5) does not bring any significant results, which casts a doubt on its added value.
- this question also raises the very important issue of quality control, as slightly suggested in the discussion (line 287). FFPE derived RNA may be of poor quality, and QC and analytical results are not presented, nor even discussed. These data must be reported in order to ensure the validity of the results
- for the Zurich cohort, there are only 11 pairs of interest, and not 31 as suggested in the abstract. This should be clearly stated
- Beyond that, tumors from the Zurich cohort are not really metastases, but rather refer to locoregional relapses. This clinically very meaningful difference may profoundly impact the results, and this should be clarified and discussed.
- the comparison between "tumors that metastasized and those who did not" is not relevant. this difference might be due only to follow up, histopathological differences, treatments, etc, and not to supposed genomic difference